# Micro-Evolutionary Processes in *Armeria maritima* at Metalliferous Sites

**DOI:** 10.3390/ijms24054650

**Published:** 2023-02-28

**Authors:** Małgorzata Wierzbicka, Agnieszka Abratowska, Olga Bemowska-Kałabun, Dorota Panufnik-Mędrzycka, Paweł Wąsowicz, Monika Wróbel, Damian Trzybiński, Krzysztof Woźniak

**Affiliations:** 1Faculty of Biology, University of Warsaw, Miecznikowa 1, 02-096 Warsaw, Poland; 2Icelandic Institute of Natural History, Borgir vid Nordurslod, 600 Akureyri, Iceland; 3Biological and Chemical Research Centre, University of Warsaw, Żwirki I Wigury 101, 02-089 Warsaw, Poland; 4Faculty of Chemistry, University of Warsaw, Pasteura 1, 02-093 Warsaw, Poland

**Keywords:** *Armeria maritima* (Mill.) Willd, heavy metals, metalliferous areas, metallophyte, microevolution

## Abstract

Tolerance to heavy metals in plants is a model process used to study adaptations to extremely unfavorable environments. One species capable of colonizing areas with high contents of heavy metals is *Armeria maritima* (Mill.) Wild. *A. maritima* plants growing in metalliferous areas differ in their morphological features and tolerance levels to heavy metals compared to individuals of the same species growing in non-metalliferous areas. The *A. maritima* adaptations to heavy metals occur at the organismal, tissue, and cellular levels (e.g., the retention of metals in roots, enrichment of the oldest leaves with metals, accumulation of metals in trichomes, and excretion of metals by salt glands of leaf epidermis). This species also undergoes physiological and biochemical adaptations (e.g., the accumulation of metals in vacuoles of the root’s tannic cells and secretion of such compounds as glutathione, organic acids, or HSP17). This work reviews the current knowledge on *A. maritima* adaptations to heavy metals occurring in zinc–lead waste heaps and the species’ genetic variation from exposure to such habitats. *A. maritima* is an excellent example of microevolution processes in plants inhabiting anthropogenically changed areas.

## 1. Introduction

Areas changed by human activity allow the tracking of the adaptations of various organisms to harsh environmental conditions such as pollution [1,2,3,4,5]. Areas intensely polluted with heavy metals found around the world are good examples. One of the industries involved is the zinc–lead ore mining and metallurgy industry, which brings about many negative environmental consequences. One effect of this activity is the formation of waste heaps of rocky debris. These sites are extremely unfavorable for plant growth, being overdried, highly insolated, poor in nutrients, and with an enormous total content of toxic heavy metals. At the same time, these areas provide scientists with an excellent opportunity to study the adaptations of organisms to live in strongly polluted environments [6,7,8,9,10,11].

To understand what these unfavorable conditions mean for plant growth, it is worth comparing them across different metalliferous areas such as those of Poland, France, and Belgium. In the Olkusz ore-bearing region in southern Poland, where many zinc–lead (calamine) waste heaps are located (e.g., the over 130-year-old zinc–lead waste heap in Bolesław), the average concentrations of heavy metals in the tested substrates (mainly calamine) were 9764 mg/kg for zinc (maximum of 72,089 mg/kg), 3657 mg/kg lead (maximum of 33,178 mg/kg), 76 mg/kg for cadmium (maximum of 506 mg/kg), and 43 mg/kg for thallium [10,12,13,14]. These amounts of heavy metals are highly abnormal compared with the geochemical background of uncontaminated soils around Poland, e.g., 5–59 mg/kg for zinc, 0.5–21 mg/kg for lead, 0.03–1 mg/kg for cadmium [15]. At the metalliferous site near a metal smelter in northern France, the concentrations of the same heavy metals in the soil were also high (0–20 cm soil layers) at 10–7140 mg/kg for zinc, 16–9920 mg/kg for lead, 23–197 mg/kg for cadmium, and additionally nearly 2000 mg/kg for copper [16]. In the substrates of the zinc–lead waste heap in the former Pb-Zn mining area in Plombières, Belgium, the zinc concentrations varied from 859 to 5000 mg/kg, lead from 186 to 2790 mg/kg, cadmium from 2.85 to 59.4 mg/kg, and copper from 13.2 to 306 mg/kg [17].

In waste heap soil (5–10 cm) in southern Poland (the Olkusz region), the contents of alkaline metals were also high (133,000 mg/kg calcium, 37,500 mg/kg magnesium), and the soil pH was neutral (~7.3) or slightly alkaline (>7.4) [14,18,19,20]. Dahmani-Muller et al. (2000) [16], after Balabane et al. (1999) [21], reported that the soil under the metallophyte grassland in northern France was moderately drained, and the rooting zone was slightly acidic (pH 5.5–6.0), with a sandy loam texture (≈60% sand, 30% silt, and 10% clay). In Plombières, Belgium, the substrate pH values ranged from 4.3 to 7.4 [17]. Metal bioavailability is considered the most important property in regulating the contents of free metal forms in the soil [22]. For example, the slightly neutral pH could be responsible for the lower metal bioavailability in the waste heap substrate in Bolesław (Poland) compared to the substrate in Plombières (Belgium), with lower pH and higher bioavailability of metals. Consequently, the calamine substrate from Bolesław showed lower toxicity compared to the soil from the waste heap in Plombières, despite the higher total metal content [17,23]. It should also be stressed that plants inhabiting metalliferous areas, in addition to adaptations to high levels of heavy metals, develop adaptations to other factors hindering growth, such as drought, strong insolation, or nutrient deficiencies associated with small amounts of organic matter in the substrate [8].

The habitats in metalliferous sites such as the zinc–lead waste heaps are extremely unfavorable for organisms; however, various taxa of vascular plants, mosses, and lichens thrive in these unusual conditions [8,9,15,24,25,26,27,28,29,30]. A common feature of these organisms is their ability to live on metalliferous soil without any symptoms of stress. They are called metallophytes. Among them are species associated only with metalliferous soils called absolute or obligatory metallophytes. They are generally highly specialized endemics that have been evolving on substrates rich in weathered ore minerals for thousands of years. Other metallophyte species occur on metalliferous and unpolluted soils and are called facultative metallophytes or pseudometallophytes. Despite their relatively short contact with metals, these organisms also exhibit many adaptive features that allow them to survive and even thrive in metalliferous areas [8,14,30,31,32,33,34,35,36].

Harsh conditions such as those described above subject plants to strong, long-term selection pressures. As a result of multi-directional adaptation, differences in morphological traits, flowering times, and tolerance degrees to the increased heavy metal contents in the substrate may occur between individuals of the same species found in metalliferous and non-metalliferous areas [8,9,10,14,26,29,35,37,38]. Such changes were observed in many metallophyte species inhabiting the zinc–lead waste heaps, e.g., *Anthoxanthum odoratum* L. [39], *Arabidopsis arenosa* (L.) Lawalrée (*Cardaminopsis arenosa* (L.) Hayek) [19], *Armeria maritima* (Mill.) Willd. [23,40,41,42,43,44,45,46,47], *Biscutella laevigata* L. [10,14,20,36,38,48,49,50,51], *Dianthus carthusianorum* L. [52,53,54,55], *Noccaea caerulescens* (J. Presl and C. Presl) F.K.Mey (*Thlaspi caerulescens* J. Presl and C. Presl) [56,57], *Silene vulgaris* (Moench) Garcke [58] or *Viola lutea* Huds., *Viola reichenbachiana* Jord. ex Boreau, *Viola riviniana* Rchb., and *Viola tricolor* L. [59,60,61,62,63,64,65,66]. Over time, microevolutionary changes may occur, leading to genetically preserved adaptations in plant populations. Their dissimilarity is sometimes so significant that it becomes a criterion for distinguishing new taxa found exclusively in metalliferous areas, e.g., *Thlaspi calaminare, Viola guestphalica*, *Viola lutea* subsp. *westfalica*, *V. lutea* subsp. *calaminaria, Minuartia verna* subsp. *hercynica*, *Biscutella laevigata* subsp. *woycickii,* and *Armeria maritima* subsp. *halleri* [3,6,7,8,10,11,14,34,36,38,56,59,67,68]. Among the species capable of colonizing areas with high contents of heavy metals, *Armeria maritima* (Mill.) Willd (sea thrift) is one of the most interesting. *A. maritima* plants growing in metalliferous areas differ in their morphological features, tolerance level, and development degree of resistance mechanisms to heavy metals compared to individuals of the same species growing in non-metalliferous areas. When inhabiting metalliferous areas, this species exhibits many adaptations to the increased contents of heavy metals in the substrate [23,26,29,40,41,42,43,44,45,46,47]. In this work, we review the current knowledge on the heavy metal adaptations of *A. maritima* found on zinc–lead waste heaps and the resulting genetic variation in this species. *A. maritima* is a good example of microevolution processes in plants that grow in anthropogenically changed areas. By focusing on *A. maritima*, a metallophyte, this work differs from other review papers on heavy metals and plants, which tend to be more general and broad.

## 2. Biology and Ecology of *Armeria maritima*

*A. maritima* belongs to the *Armeria* Willd. genus and the *Plumbaginaceae* L. family. The *Armeria* genus has about 164 species spread across all continents except Australia and Antarctica. There are several dozen species of the *Armeria* genus in Europe. The homeland of *Armeria* in Europe is the Mediterranean region [69,70]. The *A. maritima* species is extraordinarily variable, as demonstrated by its morphology. Even though the subspecies are discerned based on their characteristic traits, their differences are often blurry, which will be discussed in more detail later. The individual subspecies also differ in their geographical distribution and habitat preferences [23,69,70,71,72].

*A. maritima* is a perennial herb, classified by the Raunkiaer’s system as a hemicryptophyte—an earth-bud plant whose perennating buds are formed just below the soil surface. This species is characterized by single, evenly narrow, elongated, pointy, or blunt leaves about 2 mm wide. The leaves are gathered in a basal rosette, and an erect leafless shoot (scape) bears a single head-shaped inflorescence. The leaves are slightly pubescent, especially around the edges (less often glabrous). The tap root can reach 1.5 m in length and can have many small divaricate roots in the upper 20–30 cm of soil. The scape is 30–50 cm long, and slightly pubescent or glabrous. An individual usually has several inflorescence shoots (Figure 1a). On average, the inflorescences are approximately 12 mm in diameter. They are composed of radiant five-fold flowers, colored in shades of pink (from almost white to occasionally deep purple-red) (Figure 1b). At the base of the inflorescence head, there is a tubular scarious sheath. The inflorescence bud is supported by light green, scarious outer and inner involucral bracts. Each flower has five stamens (their filaments are 5–8 mm and broadened below), joined to the base of the petals, and five styles (filiform, free to near the base, pubescent below). The flowering period lasts from May to October. *A. maritima* has a nut-like, single-seeded fruit (about 2 mm long) dehiscing transversely above or irregularly below. The fruit is enclosed in the calyx (Figure 1c), which is tubular below and extended above into a persistent scarious pleat. During fruiting, it reaches a length of 5–7.5 mm. The fruit weights range from about 0.7 to 1 mg. Some variation in the appearance and morphology of *A. maritima* plants can be observed in the subspecies, e.g., between the subsp. *elongata*, subsp. *maritima,* and subsp. *halleri*, as shown in Table 1. Each subspecies has some unique features. However, their distinction is not always possible due to the high morphological plasticity [46,69,71,72,73,74,75].

The reproductive biology of *A. maritima* has been the subject of many studies. *A. maritima* individuals grow singly or form a compact turf by producing daughter rosettes on a joint vegetative shoot [23]. According to Lefèbvre (1976) [77], the heteromorphic self-incompatibility (due to dimorphic surface features of the pistil and pollen grains) occurring in *A. maritima* causes its strict allogamy. Eisikowitch and Woodell (1975) [78] also claimed that *A. maritima* is a dimorphic self-incompatible perennial plant. However, there are contradictory reports on the possibility of self-fertilization in this species. Philipp et al. (1992) [79] have not found any evidence of this phenomenon, whereas others [77,80,81,82,83] have recorded self-compliant (self-pollinating) *A. maritima* individuals among its metallicolous populations. According to Lefèbvre (1976) [77], the possibility of self-pollination in *A. maritima* may depend on its origin and even the way it is pollinated. Here, we question what pollinators does *A. maritima* have? For example, Eisikowitch and Woodell (1975) [78] indicated *Bombus lucorum* and *B. terrestris* as the primary pollinators of *A. maritima* subsp. *maritima* on dunes and shingles in Britain. In general, it is the common bee that frequently pollinates *A. maritima* [78].

*A. maritima* inhabits much of the northern hemisphere. The species is common across most of continental Europe, except for the eastern region. The range of *A. maritima* subsp. *maritima*, an obligatory halophyte occurring only in saline areas, includes the Baltic, North Sea, and Atlantic coasts, is a dominant subspecies in Western Europe. Less commonly distributed is *A. maritima* subsp. *elongata,* with its range stretching over Central and Eastern Europe. *A. maritima* subsp. *halleri,* with its disjunctive range covering only metalliferous regions, is the taxon spanning the narrowest range. This subspecies is an obligatory metallophyte and is regarded as an indicator of copper, zinc, and lead ore-bearing areas. It is common in the regions where non-ferrous metal ores occur and are processed (Table 1) [23,47,70,72,76]. *A. maritima* occurs in grassland communities, dry grasslands on sandy soils, as well as areas with specific habitat conditions (mountain, saline, and metalliferous regions). Generally, the species prefers mineral-humus, fresh, and moderately poor (mesotrophic) soils with neutral pH (about 6–7), which corresponds with the conditions found on waste heaps in three metalliferous areas (Poland, France, and Belgium), as described in the previous chapter. The species occurs in such phytocoenoses as *Vicio lathyroidis-Potentillion argenteae*, *Diantho-Armerietum elongate*, and *Corynephoro-Silenetum tataricae* [16,17,23,69,70,72,74,76,84]. *A. maritima* is characterized by a broad ecological amplitude and may settle in natural wild or extremely harsh environments, such as the zinc–lead waste heaps. Given the robust nature of this species, the adaptations *A. maritima* has developed to live in metalliferous areas are worth examining. 

### 2.1. Armeria maritima Adaptations to Heavy Metals

To accurately study the adaptation mechanisms in *A. maritima*, the research was conducted under controlled laboratory conditions. Multiple research techniques—including morphology, cytology, electron microscopy, and molecular biology—were used to perform experiments and compare results. The research on the *A. maritima* adaptations to heavy-metal-enriched habitats has revealed a complex network of mechanisms working in unison to keep the plant in good condition. Studies show that *A. maritima* specimens from metalliferous areas differ in their morphological features, tolerance levels, and development degrees of resistance mechanisms to heavy metals compared to their counterparts growing in non-metalliferous areas. For example, Olko et al. (2008) [46] and Abratowska et al. (2015) [47] showed that *A. maritima* populations from a metalliferous area (zinc–lead waste heap in Bolesław, Poland) and non-metalliferous area (a rural area in southern Poland) differed in their tolerance to zinc, lead, and cadmium. Plants from the metalliferous population showed a higher tolerance to these metals by about 20% in the Wilkins’ root tolerance tests compared to plants from the non-metalliferous area. In this section, the most advanced adaptations of *A. maritima* to heavy metals will be discussed, and the differences in metal protective responses resulting from a microevolution process between plants from metalliferous and non-metalliferous areas will be presented.

### 2.2. Avoidance and Tolerance Mechanisms

Before discussing *A. maritima’s* adaptations to growth in a metalliferous area, it is essential to differentiate the general protective responses of plants to heavy metals. These mechanisms are divided into two groups: (1) avoidance of heavy metals, which prevents metals entering ions into a cell; (2) tolerance of heavy metals, which is a response to the presence of metal ions in a cell [14,30,45,85]. 

Plants deploy the following avoidance strategies:Exclusion—processes that prevent the absorption of heavy metals by a plant, e.g., the release of metal-ion-chelating compounds into the rhizosphere, via which metals become immobilized or captured in roots and shoots, preventing the spread of metals in a plant;Elimination—processes that work after heavy metals have entered a plant’s tissues and lead to the expulsion of metals to the outside, e.g., heavy metals are excreted through the plant surface, secreted by glands and trichomes, or the entire organs with the heavy metal content, such as the oldest leaves, wither and fall off;Redistribution—processes that reduce the presence of heavy metals and transport them to sections of the plant, where there is a lower risk of their toxic effects on an organism, such as aging leaves; this process works in combination with elimination;Compartmentation—a process that occurs at the cellular level, wherein heavy metals accumulate in regions of the plant where they no longer threaten the cells, e.g., in cell walls, intercellular spaces, and vacuoles.

In turn, the tolerance mechanisms work when heavy metal ions have already entered the cell and pose a threat to cellular metabolism. In response to an increase in metal concentration in the cytoplasm, stress proteins (also synthesized in response to other stressors), polypeptides, and stress metabolites are secreted, e.g., heat shock proteins (HSP) and ras-associated binding proteins (RAB), osmotins, proline, metal chelators such as glutathione or phytochelatin, amino and organic acids, signal transducers, structural proteins, enzymes, and many others [14,30,35,37,45,58,85,86,87,88,89,90].

### 2.3. A. maritima Adaptation to Heavy Metals at the Levels of the Whole Organism, Individual Tissues, and Cells

The most important resistance strategy of *A. maritima* at the organism level was the retention of heavy metals (lead, cadmium, zinc, and copper) in the roots and the metal accumulation in the oldest withering leaves, as rejected by the plants (redistribution and elimination, also associated with compartmentation and exclusion). This mechanism limits the transport of metals to green leaves and generative organs. Interestingly, these strategies are common to plants growing in contaminated areas (generation F0) and those cultivated with heavy metals in laboratory conditions (generation F1). *A. maritima* plants cultivated experimentally in mineral media were characterized by a similar distribution of metals as plants growing in outdoor conditions in metalliferous areas. This observation indicates the genetic fixation of the ability to defend against heavy metals in *A. maritima* plants from metalliferous sites. Moreover, cultivated under laboratory conditions, plants from metalliferous and non-metalliferous populations showed the same metal accumulation pattern, indicating that this is a feature inherent to the *A. maritima* species [15,16,23,35,39,40,41,45,46]. The subsequent sections describe these protective responses in more detail.

The research on *A. maritima* showed zinc accumulation in the outer layers of the root, more specifically the rhizoderma and cortical cells, concentrated mainly in the cell walls and vacuoles (compartmentation mechanism) [41,45]. Heavy metals are transported in these layers by the apoplastic route up to the innermost cortex, the endoderm. The endoderm is a single layer of cells. Suberinic thickening, the so-called Caspary’s strand, surrounds each of those cells and stops the apoplastic radial transport. The only way substances may penetrate further is via the symplast of the endodermal cells. This suberinic barrier is very effective in blocking the transport of heavy metals deep into the root and preventing their movement to the aboveground parts of the plant [14,15,41,45,53]. Some research has shown that the endoderm of the *A. maritima* root retains zinc by limiting its transport to the aboveground parts [41,45]. Szarek et al. (2004) [15], who studied the metal accumulation in *A. maritima* from metalliferous and non-metalliferous soils in laboratory conditions, found significant differences in the concentrations of lead, zinc, and cadmium between the roots and green leave. This result demonstrated the limited transport of metals from the roots to aboveground parts of *A. maritima*. This is an example of an exclusion strategy [15]. Similarly, Brewin et al. (2003) [91] showed that *A. maritima* roots and living leaves generally act as copper excluders, whereas decaying leaves act as copper accumulators. The authors pointed out that the higher copper retention in the roots compared to the living leaves and copper excretion through the decaying leaves point to different mechanisms of copper tolerance [91]. The work by Dahmani-Muller et al. (2000) [16] showed that in *A. maritima* subsp. *halleri,* the lead and copper concentrations were 20 and 88 times higher, respectively, in the roots than in the green leaves; the authors suggested an exclusion strategy by metal immobilization in the roots. Relevant to this matter is the study by Neumann et al. (1995) [40], who showed that *A. maritima* subsp. *halleri* accumulates copper in the vacuoles of the tannic root cells (also known as idioblasts) via chelation with polyhydroxy phenolic compounds; this issue will be discussed in more detail in later sections of the review. Dahmani-Muller et al. (2000) [16] suggested that such metal–phenol complexes are transported from the roots to aging leaves, which brown and fall. Ernst et al. (1992) [92] also proposed leaf shedding as a metal detoxification mechanism [16].

The characteristic trait of *A. maritima* is the intensified metal accumulation in the oldest leaves compared to the green ones. Inhibiting the transport of metal ions to green leaves and generative organs protects plants’ most sensitive processes, i.e., photosynthesis and reproduction, against the toxic effects of heavy metals [15,16,23]. Dahmani-Muller et al. (2000) [16] showed that the zinc, cadmium, lead, and copper concentrations in the brown leaves of *A. maritima* subsp. *halleri* from northern France were 3–8 times higher than in green leaves. The authors linked the subsequent leaf fall to the detoxification mechanism [16]. In turn, the lead concentration in the oldest leaves of the *A. maritima* plants growing on the waste heaps in Bolesław (the Olkusz region in Poland) and the Plombières region exceeded the hyperaccumulation level, i.e., it surpassed 1000 mg/kg DW [23,47].

The plants cultivated in laboratory conditions—in mineral solutions enriched with metals—were characterized by similar metal distributions as the plants growing on waste heaps. Moreover, the experiments under laboratory conditions showed that the metal accumulation process was the same in the plants from metalliferous and non-metalliferous populations [15,23,47]. Dahmani-Muller et al. (2000) [16] suggested that the translocation of metals to senescent leaves in the *A. maritima* plants might be an active process assisted by phenolic compounds, which the tissues of *A. maritima* plants are rich in [16]. The metal quantity accumulated in the tissues of these plants was very high, exceeding the lethal level for most ‘ordinary’ plants. This accumulation pattern may be a specific trait of the A. maritima species.

How does *A. maritima* transport metal elements around its cellular body? Their transport method in leaves from the vascular bundle to the epidermis passes through the mesophyll—a tissue composed of photosynthetically active cells—is the most sensitive to the toxic effects of metals. Therefore, metal transport must involve both the apoplastic and symplastic pathways. In the *A. maritima* plants, unique mechanisms prevent the harmful effects of excess metal in mesophyll [15,23,47]. The research on lead transport in the *A. maritima* cut-off leaves showed that the main lead accumulation areas were vascular bundles, mesophyll cell walls (primarily those located close to the bundle), and intercellular spaces. Only a small amount of lead penetrated the cells. Other data showed that lead, cadmium, and zinc were found mainly in vascular bundles and apoplasts of the mesophyll area. Thus, one of the detoxification mechanisms in the mesophyll region is to bind heavy metals in cell walls and intercellular spaces (the compartmentation mechanism) [45,47]. Additionally, Heumann et al. (2002) [41] found zinc grains in the leaves of *A. maritima* subsp. *helleri*. Specifically, these grains localized extracellularly to all cell walls and intracellularly to xylem vessels, vacuoles of transfer cells, parenchyma cells, gland cells, and plasmodesmata of various cell types. In general, the zinc found in those leaves accumulated in the cell walls of vascular bundles and the cell walls and vacuoles of mesophyll cells [41]. Substantial metal accumulation was also found in and around the secretory cells, as discussed in the subsequent paragraphs. Other authors observed similar phenomena [41,45,46,47]. For example, Szarek-Łukaszewska (2004) [15] found the largest concentrations of heavy metals in withering leaves in vascular tissues and surrounding parenchymal cells. She also detected metals in the parenchyma and epidermis cell walls, coinciding with the results described in this paragraph.

Moving metals to sections of the plant where they do not pose a direct hazard to metabolism is one of the plants’ most essential detoxification mechanisms [87,93]. An interesting avoidance mechanism against heavy metals in the *A. maritima* plants relates to the functioning of the epidermal structures—the trichomes and salt glands—an example of the elimination mechanism [23,46,47]. The epidermis and its structures, like trichomes, are the target places to accumulate metals in leaves. The epidermal cells, except for the stomatal guard cells, are characterized by lower metabolic activity than the mesophyll cells. The epidermal structures may, therefore, accumulate the excess of metals, lowering their concentration in the mesophyll cells of leaves. Consequently, the epidermal structures protect sensitive physiological processes [89]. However, whether the mechanisms related to the functioning of epidemic structures are more efficient in plants from metalliferous than non-metalliferous sites awaits confirmation.

According to various authors, heavy metals such as lead, cadmium, zinc, and copper accumulate in the trichomes and salt glands of the *A. maritima* leaf epidermis [23,41,46,47]. For example, the metals accumulate in the epidermal trichomes after the cultivation of the *A. maritima* plants with the addition of lead, cadmium, and zinc. The metals localize the cell walls of trichomes, especially at their base and on the inside of the cells. Data show that *A. maritima* plants from metalliferous and non-metalliferous populations rely on this mechanism [45,47]. The study by Olko et al. (2008) [46] found the highest metal concentrations in the leaf epidermal and subepidermal tissues, including trichomes. Thus, trichomes may play an important role in heavy metal storage. Similar results, showing metal accumulation in epidermal structures, including trichomes, were also obtained for other metallophytes, including *S. vulgaris* [94], *T. caerulescens* [90,95], *A. halleri* [96,97], or *B. laevigata* [14,48]. In the cited studies, metal accumulation in the epidermal cells and trichomes was considered one of the most critical mechanisms of hyperaccumulation. The objective of other studies using different species was also to explore the mechanism of metal accumulation in trichomes. For example, in the *Vicia faba* epidermal trichome cells, the metal action increased the expression of metallothioneins [98]. On the other hand, in the *A. halleri* trichomes, there was a higher level of glutathione and increased expression of genes encoding its biosynthesis [99]. Further research on the mechanism of metal accumulation in the *A. maritima* trichomes is still needed.

Another feature of *A. maritima* is the heavy metal secretion by salt glands [23,40,41,42,43,44,46,47]. These specialized secretory structures are formed by the leaf epidermis. They play a key role in regulating salt concentrations in halophytes that grow on saline substrates. The saline solution concentrates inside the secretory cells of the gland and travels to the zone between the cellulosic part of the cell wall and the cuticle layer lifted over the gland. The solution then drains through the pores in the cuticle. The secretory cells differ in their ultrastructure from other epidermal cells. The difference, mainly in the bigger number of mitochondria, indicates that salt secretion is an active process requiring an energy supply. A thick layer of cutin that saturates the cell walls separates the secretory cells from the mesophyll cells. This barrier prevents a backflow of the concentrated salt solution to the mesophyll [100,101,102]. There is also a hydrophobic substance between the secretory cells and the internal cells at the base of the salt gland, which does not saturate the cell wall but adheres to it. As a result, a continuum of the apoplast between the mesophyll cells and the cells of the gland base is preserved [102].

Naturally, *A. maritima* often grows in highly saline soils, e.g., coastal areas. The salt glands are found in its leaves’ upper and lower epidermis. A cuticular layer surrounds them with pores at the leaf surface and cutin-free transfusion zones, where outer basal cells are contiguous with surrounding sub-basal cells. The salt glands are built from sixteen cells (arranged into four quadrants). Four secretory cells, located in the inner quadrant, are surrounded apically by four accessory cells. Four inner and four outer basal cells cover the secretory and accessory cells. All of the cells have a very dense cytoplasm rich in organelles. They also have more or less developed wall protuberances. In the *A. maritima* plants growing in saline areas, the action of salt glands is an important immune mechanism that allows the removal of excess salt from leaf tissues [41,69,74]. Does the appearance of salt glands differ between subspecies? According to Heumann (2002) [41], the upper and lower leaf epidermal tissues of *A. maritima* subsp. *halleri* possess specialized multicellular salt-secreting glands. Their structure is identical to that in the halophyte, *A. maritima* subsp. *Maritima,* and other salt-tolerant *Plumbaginaceae* [41]. In turn, Neumann et al. (1995) [40] showed that *A. maritima* subsp. *halleri* growing on non-contaminated soil displays the same morphologic structure of leaf salt glands as plants from the mine mound in the medieval copper mining region in Germany. Thus, the salt glands appear the same within the whole *A. maritima* species [40,41]. Several studies indicate that the salt glands are involved in metal detoxification in the *A. maritima* plants growing in metalliferous areas. The removal of heavy metals through salt glands is one of the species’ adaptations, but its mechanism is still not fully understood [23,40,41,42,43,44,46,47]. For example, salt crystals covered the leaves of the *A. maritima* plants during cultivation in a substrate supplemented with lead, cadmium, and zinc. The three metals given to the plants were found in the epidermal salt gland cells and the salt content secreted onto the leaf surface. They were, thus, secreted onto the leaf surface by the salt glands. The secreted solution also contained other elements and mineral components of the cultivation substrate, including calcium, sulfur, chlorine, magnesium, and potassium. Therefore, the regulation of the metal concentration in the leaf mesophyll depended on the salt glands. This mechanism occurred in plants from metalliferous (e.g., zinc–lead heaps) and non-metalliferous regions [23,44,47].

### 2.4. Armeria maritima Is a Halophyte That Adapted to Salt Excess in the Soil

*A. maritima’s* salt glands (Figure 2) secrete excessive amounts of salts onto the leaf surface (Figure 2a). During the experimental study, in which *A. maritima* plants were grown in a medium with excess salt, clusters of crystals appeared on the leaves within four weeks from the start of the cultivation (Figure 2b). An analysis using electron microscopy with X-ray microanalysis showed that the crystals included lead on their surfaces. Traces of sulfur, calcium, and phosphorus were also present in the crystals. Applying the single-crystal X-ray diffraction analysis provided insights into the structure and revealed that various chemical compounds form the crystals, such as potassium nitrate (KNO_3_) and calcium sulphate dihydrate (CaSO_4_·2H_2_O). The salt glands were highly effective at removing elemental lead. A comparison of the lead concentrations on rinsed and unrinsed leaves showed that about 40% of the lead found on the leaves was rinsed out. The following example illustrates how effective the process of removing lead from *A. maritima* plants through salt glands is. If an *A. maritima* plant contained 100 mg Pb/kg DW in its leaves and 40% of the Pb was rinsed out on the leaf surfaces, then only 60 mg Pb/kg DW would remain inside the leaf tissues [103]. The efficiency of this process is extraordinary. This mechanism of lead detoxification is specific to this plant species.

The contribution of salt glands to the detoxification of the *A. maritima* plants exposed to excess heavy metals was also observed by others [40,41]. The location of zinc in the salt glands was described in detail by Heumann (2002) [41], who used TEM and X-ray microanalyses. The author showed that compact masses of zinc deposits were localized outside the cuticle, particularly over the secretion cells. The scattered zinc granules were present in the cell walls of all gland cells and sub-basal cells, except for the cutinized layers found at the apical and basal areas of the gland. Heumann (2002) [41] also showed that the cytoplasm of the gland cells contained numerous vesicles of different sizes with dark zinc deposits. In the sub-basal cells, the zinc deposits found in the large vacuole were always adjacent to the tonoplast. Plasmodesmata were observed between all neighboring cells described above, but zinc deposits were found only in plasmodesmata between sub-basal and outer basal cells [41]. In turn, Neumann et al. (1995) [40] studied copper secretion by the *A. maritima* salt glands. According to the authors, copper ions reaching the vascular bundle were translocated via the transpiration stream into the leaves. They were excreted partly by the salt glands on both leaf surfaces. Solutes excreted by the salt glands formed numerous small crystals on the leaf surface. Moreover, the crystals contained phosphorus, sulfur, chlorine, potassium calcium, and large amounts of copper, as well as traces of zinc, nickel, iron, and manganese [40]. The elemental content of the crystals on the leaf surface in *A. maritima*, found by Neumann et al. (1995) [40], was confirmed in the other already cited reports [23,44,47]. However, chemical forms of the metals secreted by the *A. maritima* salt glands have not been identified yet. These might be calcium, magnesium, potassium, lead, cadmium, zinc chlorides, sulphates, or carbonates. However, they may also be complex inorganic salts containing more than one metal. Therefore, metal secretion by the *A. maritima* salt glands still requires further research.

Finally, it is worth mentioning that this type of metal detoxification mechanism has also been described in other plant species, e.g., water lily (*Nymphaea*), in which the salt glands secrete salts containing lead, cadmium, and mercury in combination with other ions, mainly calcium [104,105,106]. Undoubtedly, in the *A. maritima* plants, removing heavy metals from their tissues is one of the protective responses enabling them to grow and develop in metal-rich soil.

### 2.5. A. maritima Adaptation to Heavy Metals at a Physiological and Biochemical Level

The production of glutathione (GSH), phytochelatins (PCs), and organic acids in plants is a manifestation of tolerance mechanisms. These compounds maintain intracellular homeostasis by detoxifying metal ions and protecting against their toxic effects. Glutathione is an antioxidant. An increase in its amount protects cells against oxidative stress. Phytochelatins formed from glutathione with the phytochelatin synthase activity bind free metal forms within the cytosol and participate in intracellular metal transport. Organic acids participate in the long-distance transport of metals and bind metals within a vacuole, playing a detoxification function inside a cell [14,15,46,107,108,109,110,111].

Growth adaptations under excess toxic heavy metals were also investigated at the physiological and biochemical levels of the *A. maritima* plants. No phytochelatins were found in the *A. maritima* plants cultivated with zinc, lead, and cadmium [46]. However, the authors of the study observed an increase in the glutathione content and changes in the content and proportions of organic acids, such as malic acid. The glutathione pool increased in the plants from the zinc–lead waste heap in Bolesław (near Olkusz in Poland) and in those from the non-contaminated area. Additionally, they found a change in the malic acid levels in plants from the zinc–lead waste heap population. The contents decreased in the roots but increased significantly in the leaves. Such a high increase in the malic acid content in the leaves indicated its intensified transport and the synthesis of additional pools in those leaves [23,46,47].

Another tolerance mechanism in the *A. maritima* plants is related to the presence of the so-called tannin cells containing phenolic compounds in vacuoles. These cells occur abundantly throughout the tissues of this species. In the *A. maritima* plants growing on soils enriched with zinc and copper, the tannin cells accumulate metals [40,41]. Neumann et al. (1995) [40] showed that in the roots of *A. maritima,* the multiseriated sclerenchymatous exodermis contains various types of idioblasts. In the cortical parenchyma, idioblasts (which included the vacuolar clusters and osmiophilic material between the cell walls and the plasmalemma) were often found. This work also showed that many idioblasts with osmiophilic precipitates in the vacuoles occur in the leaves, their epidermis, the palisade parenchyma, and the spongy parenchyma—around and inside the bundle. A similar structure was found in plants from metalliferous (a medieval copper mining region in Saugrund-Eisleben, Germany) and non-metalliferous areas (the Botanical Gardens, Halle, Germany). In the roots of *A. maritima,* the vacuoles of numerous idioblasts turned out to be the primary storage compartments for copper [40]. Other studies showed that during plant cultivation in the lead-, cadmium-, and zinc-enriched medium, phenolic compounds in the vacuoles of the root tannin cells frequently bound the heavy metals [23,47]. The *A. maritima* population from the metalliferous area (a waste heap in Bolesław, Poland) showed a higher tolerance level to lead, cadmium, and zinc than the population from the non-contaminated site. In the roots of *A. maritima* from the contaminated area, the number of tannin cells was higher than in the more heavy-metal-sensitive population from the uncontaminated area. The vacuoles of these cells contained phenolic compounds with heavy metal deposits. The deposits comprised lead, cadmium, or zinc, depending on which metals were given to the plants during growth. Therefore, the protective action of phenolic compounds was a critical feature responsible for increasing the tolerance level to the metals in the *A. maritima* plants from zinc–lead waste heaps [23,47]. For comparison, Heumann (2002) [41] found in the roots of *A. maritima* from the metalliferous area (near a lead smelter at Stolberg, Germany) zinc-enriched granules—extracellularly in the cell walls of the rhizodermal and outer cortical cells, all the way to the anticlinal walls of the endodermis. Similar deposits were also found intracellularly in the vacuoles of rhizodermal cells, the outer cortical cells, the endodermis cells, and the xylem vessels. The zinc granules were also found in leaves extracellularly in all cell walls; intracellularly in the xylem vessels, vacuoles of transfer cells, parenchyma, and gland cells; and in plasmodesmata between distinct cell types [41]. Therefore, in addition to the vacuoles, the cell walls are also important sites for metal detoxification in this species.

The last physiological–biochemical adaptation to the excess of heavy metals was found in the *A. maritima* plants growing in the region of the former copper mine in Germany. The cytosol of the root cells of these plants contains the HSP17 low molecular weight stress protein. In contrast, HSP17 was not found in the leaves [40]. HSP17 belongs to the heat shock proteins, formed not only at high temperatures but also engaging in heavy metal tolerance. They participate in the repair of intracellular damage caused by metals. HSPs function as chaperones in protein folding and assembly and can also protect and repair proteins damaged by oxidative stress caused by metals [14,30,40,47,112].

## 3. Variation within the *Armeria maritima* Species from the Metalliferous and Non-Metalliferous Areas, Microevolution Aspects—Existing Knowledge

As mentioned in the introduction, the microevolutionary process in plants from metalliferous areas is a common phenomenon. Microevolution leads to the emergence of genetically fixed differences within a species. This phenomenon involves evolution at the population level, which concerns changes in time in the frequency of alleles in a given population. These changes result from mutation, natural selection, genetic drift, or gene flow. Presumably, epigenetic agents also play a significant role in this process. Microevolution leads to interpopulation variability of the taxon. The exemplary results of microevolution are the above-described adaptations of *A. maritima* to grow in metalliferous areas. New adaptations are hereditary and increase the likelihood of survival and reproduction in a specific environment, such as an area contaminated with heavy metals. It is also worth noting that the mechanisms of microevolution in plant populations in areas contaminated with heavy metals, such as zinc–lead waste heaps, are not fully elucidated, and many studies continue to raise this question [2,3,4,5,7,8,9,11,14,30].

Noteworthy is also the fact that due to increasing environmental pollution, plant microevolution processes in degraded habitats such as metalliferous areas become particularly important, because over time they may lead to the formation of new taxa; that is, new forms, varieties, and even subspecies [3,7,8,9,10,11,14,38]. The microevolution process may influence the differentiation of the *A. maritima* populations on soils highly contaminated with heavy metals. However, the taxonomic approach to this species is highly complicated, as stated explicitly in this chapter.

As mentioned above, within *A. maritima* several subspecies have been distinguished, differing in terms of their morphological features, geographic distribution, and occupied habitat (Figure 3; Table 1). The *A. maritima* species is characterized by high phenotypic plasticity related to habitat conditions [23,47,74]. In the studies by Pinto da Silva (1972) [69] and Baumbach (2012) [113], one can find information that *A. maritima* is a highly polymorphic species. The confusion about the taxa of *A. maritima* was so great that the subspecies were distinguished many times before lowering their rank to that of a variety. In general, the two subspecies of *A. maritima*, *elongata* and *maritima,* are at present distinguished in non-metalliferous areas, while the subsp. *halleri,* according to most authors, occurs in metalliferous areas. The following section deals mainly with the third subspecies.

The populations of *A. maritima* subsp. *halleri* in metalliferous localities have been the subject of botanists’ interest dating back to the 19th century. In 1844, a population growing in the area of the Upper Hartz Mountains (Germany) was distinguished from the species. It was accepted first as an endemic species, *Armeria halleri*. However, later it was ranged as a subspecies, *Armeria maritima* (Mill.) Willd. subsp. *halleri* (Wallr.) Á. Löve and D. Löve s. str. In later years, this subspecies also included several micro-endemic populations in Western and Central Europe’s metalliferous (zinc–lead and copper) areas. Currently, the populations of all zinc–lead and copper localities of Western and Central Europe are ranked into *A. maritima* subsp. *halleri* s. l., characterized by a discontinuous range determined by the locations of metalliferous areas. The taxon *A. maritima* subsp. *halleri* s. l. includes the populations previously classified as *A. maritima* subsp. *halleri* (Wallr.) Á. Löve and D. Löve; *A. maritima* subsp. *calaminaria* (F. Petri) W. Ernst; *A. maritima* subsp. *eifeliaca* (F. Petri) Lefèbvre; *A. maritima* subsp. *hornburgensis* (A. Schulz) Rothm., and *A. maritima* subsp. *bottendorfensis* (A. Schulz) Rothm [23,43,47,69,76,113,114,115,116,117].

The researchers who dealt with the systematics of *A. maritima* around metalliferous sites in different parts of Europe highlighted significant variability in traits between populations. Lefèbvre (1974) [76] showed that the morphotype of individuals in populations from metalliferous soils always made it possible to distinguish them from the plants from non-metalliferous soils. However, it is difficult to characterize the morphology of A. maritima subsp. *halleri* unequivocally due to its polymorphism [67,76]. The lack of straightforward morphological characteristics of the populations from metalliferous areas has also been repeatedly pointed out. Still, some diagnostic features allow the distinction of *A. maritima* subsp. *halleri* from subsp. *elongata* and subsp. *maritima.* The most important features are as follows: (1) the length of the generative shoots do not exceed 30 cm (they are taller than *A. maritima* subsp. *maritima*, but shorter than in subsp. *elongata*); (2) the outer involucral bracts of the inflorescence bud are not longer than the bud (in contrast to other subspecies, in which they can be even twice as long as the bud); (3) the color of the flowers is dark pink or purple-pink (in other subspecies the corolla is usually light pink), as shown in Table 1 [23,47,67,69,76].

The studies performed with isozymes [118,119] and RAPD (random amplification of polymorphic DNA) markers [42] showed that the large morphological variation corresponds to the large genetic variation occurring within the *A. maritima* species and strong differentiation at the population level, as pointed out by Baumbach (2012) [113] in his review article. In light of this, the classification of its metalliferous populations is challenging. The reason is that *A. maritima* subsp. *halleri* evolves from populations occurring at the nearest non-metalliferous area and belonging to *A. maritima* subsp. *elongata* or *A. maritima* subsp. *maritima,* which in turn are determined by the geographical location. Accordingly, the *A. maritima* individuals in different metalliferous regions may be more similar to *A. maritima* subsp. *maritima,* the subspecies dominating in Western Europe, or *A. maritima* subsp. *elongata, which is* common in Central and Eastern Europe. This distinction was indicated, for example, by genetic tests performed with the AFLP (amplified fragment length polymorphism) method on the *A. maritima* plants found in Germany [43,113]. These studies give evidence that metalliferous populations evolved from ancestral non-metalliferous populations repeatedly and independently in different geographical regions. As in the previous studies using phenotypic markers, molecular studies showed the lack of a habitat-related genetic structure of the *A. maritima* populations. Thus, the conclusion was that all studied populations from the Central European metalliferous sites are edaphic varieties of the subspecies *A. maritima* subsp. *elongata* [43,113]. The studies of Western European populations, using variability markers such as the enzymatic profiles, pollen size, ploidy, or DNA content, did not allow the taxonomic units of *A. maritima* in metalliferous and non-metalliferous sites to be distinguished either [82,118,119,120,121,122,123]. It is worth adding that because subsp. *halleri* cannot be consistently characterized throughout its geographic range and may be an artefact itself, Baumbach and Hellwig (2007) [43] and Baumbach (2012) [113] postulated that all endemic forms from metalliferous areas (e.g., ‘hornburgensis,’ ‘bottendorfensis,’ ‘eifeliaca,’ or ‘calaminaria’) should be treated as varieties of *A. maritima* subsp. *elongata*.

To sum up, the studies cited showed that the most frequent similarity of the different *A. maritima* populations did not result from the type of occupied habitat but the geographical location. The populations at the various metalliferous sites were less similar than the geographically closest populations at the non-metalliferous sites. This confirmed the polyphyletic genesis of *A. maritima* subsp. *halleri* that occurs in metalliferous areas.

In turn, Abratowska et al. (2012, 2015) [23,47] used a variant of the AFLP technique based on restriction enzymes with different sensitivity levels to DNA methylation (metAFLP). This approach allows for the simultaneous comparison of polymorphisms at the DNA sequence level and the methylation pattern level. The latter informs about the variability at the epigenetic level. The cited studies compared two metalliferous populations (zinc–lead waste heap from Bolesław in Poland and zinc–lead slag heap from Plombières in Belgium) and four non-metalliferous populations (dry and semi-dry meadow, Poland) (Figure 4). The data showed that the *A. maritima* plants from the metalliferous area in Bolesław (Poland) differ from both the population from the metalliferous area in Plombières (Belgium) and the non-metalliferous sites in south-eastern Poland. They constitute a genetically separate group [23,47]. These studies also showed that the pattern of the genetic structure in the populations investigated changed considerably in relation to the pattern of DNA methylation (Figure 4). Due to the value of the genetic distance, the population from the metalliferous area in Poland was closer to the metalliferous population from Belgium than to the population from the non-metalliferous area in Poland. Therefore, in terms of the DNA methylation pattern, the plants from metalliferous regions were more similar than those from non-metalliferous areas. These studies showed the naturally occurring epigenetic variation in the *A. maritima* populations, which may result from the microevolutionary process in the metalliferous regions [23,47]. The above considerations indicate that the mechanisms of microevolution in plant populations in metalliferous areas remain incompletely elucidated. Presumably, epigenetic agents play a significant role in this process. Modifying the DNA methylation pattern in plants under stress conditions impacts the regulation of gene expression [10,124,125,126]. The results of the studies using molecular markers [23,47] suggest that this kind of regulation may influence the differentiation of the *A. maritima* populations on soils highly contaminated with heavy metals.

## 4. Future Research

The molecular adaptation mechanisms of plants counteracting the heavy metals in soil is a separate issue of sizable proportions that has already been summarized in review papers and books. Many studies on plants and heavy metals have already been published. They concern such metals as Fe, Mn, Cu, Ni, Co, Cd, Hg, and As [127,128]. The authors indicate that as a line of defense, other mechanisms for detoxification of these metals are introduced that transport, chelate, sequester, and detoxify these metal ions in the plant’s vacuole [127,129]. A molecular study on the mechanisms of heavy metal tolerance in plants was also thoroughly described in a paper by Ding et al. (2020) [130] on the role of microRNAs in heavy metal tolerance. The book Plants and Heavy Metals [129] comprehensively describes plant defense mechanisms against heavy metals. For example, the author points out that many plants exposed to toxic concentrations of metal ions reduce the uptake into their root cells by restricting metal ions to the apoplast, binding them to the cell wall or cellular exudates, or by inhibiting long-distance transport. These processes result in changes in cell membranes that impact membrane transport in particular. However, such studies on *A. maritima* plants are lacking.

In this work, we have presented two populations of *A. maritima* from metal-rich and non-metal-rich soils found in Poland. The current genetic studies show that these populations are dissimilar. This means that plants from metal-bearing areas near Olkusz demonstrate higher tolerance compared to the non-metal-bearing population. The present work is an excellent material for future comparative studies using molecular methods or experiments to enrich plants with genes responsible for high tolerance to heavy metals.

We also indicate that anthropogenic areas with high levels of heavy metals, where interesting varieties and subspecies of plants may occur, need to be recognized and protected.

## 5. Conclusions

The present work shows that the resistance of *A. maritima* plants to heavy metals depends on a complex network of interrelated processes, with examples listed below:The retention in the roots and accumulation in the oldest leaves of metals that then dry up and undergo abscission—limiting the concentration of metals in green leaves and generative shoots and protecting photosynthesis, flower, and seed development;The binding of metals in the cell walls (roots)—preventing the penetration of metals into the cell and protecting them from disturbances in ultrastructure and metabolic processes in the cytosol;The complexation of metal ions by phenolic compounds and via sequestration in the vacuoles (roots and leaves)—regulating the metal concentration in the cytosol and protecting against the toxic effects of metals;Metal deposition in epidermal trichomes—regulating the metal concentration in the leaf mesophyll and protecting photosynthesis;Metal secretion by epidermal salt glands—regulating the metal concentration in the leaf mesophyll and protecting photosynthesis.

In *A. maritima,* the phenotypic and genetic variations are significant, as indicated by numerous studies. The genetic studies discussed here show that in different metalliferous populations of *A. maritima,* the increased heavy metal tolerance mechanisms due to microevolution developed independently of many iterations compared to their close non-metalliferous populations. Nevertheless, the considerations presented here indicate that genetic studies on the *A. maritima* populations from metalliferous sites are still necessary, using other methods such as microsatellites, SNPs (single-nucleotide polymorphisms), RFLPs (restriction fragment length polymorphisms), or DNA sequencing, because the past research on this subject has not fully elucidated the observed phenomena.

## Figures and Tables

**Figure 1 ijms-24-04650-f001:**
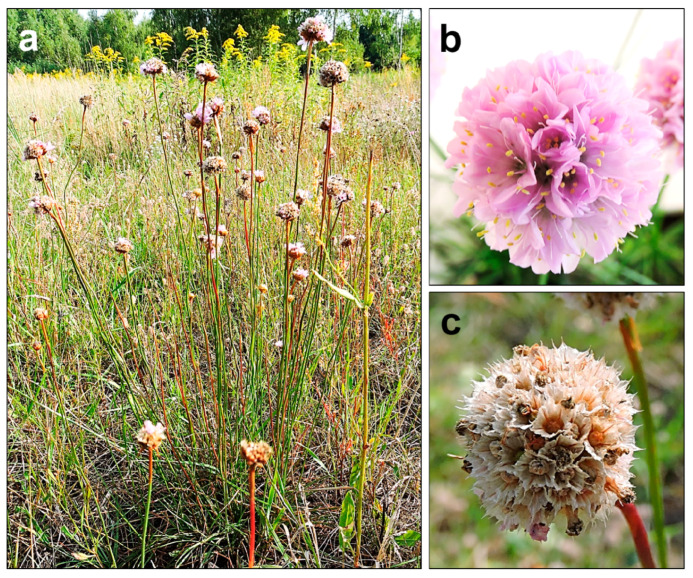
The appearance of *A. maritima* plants from the area near Warsaw (Poland), not contaminated with heavy metals: (**a**) habitat of the whole plant; (**b**) inflorescence with developed flowers; (**c**) seeds in calyxes. All photos by Olga Bemowska-Kałabun.

**Figure 2 ijms-24-04650-f002:**
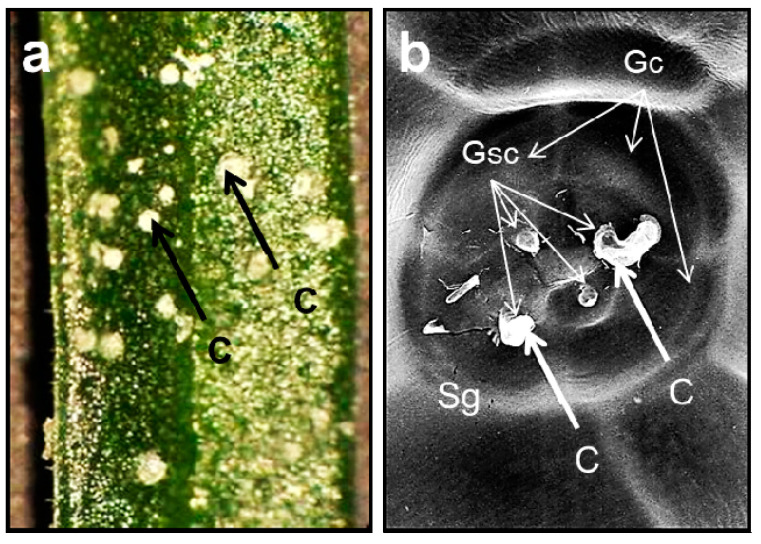
Leaf area of *A. maritima* after four months of culture in medium supplemented with 20 mM lead nitrate: (**a**) crystals on the leaf, magnification 25×; (**b**) salt glands among the cells of the epidermis cells; scanning electron microscope, magnification 2000×. Abbreviations: C—salt crystal; Gc—cells of the gland; Gsc—secretory cells of the gland; Sg—salt gland. All photos by Agnieszka Abratowska (based on Abratowska et al., 2015 [47], modified).

**Figure 3 ijms-24-04650-f003:**
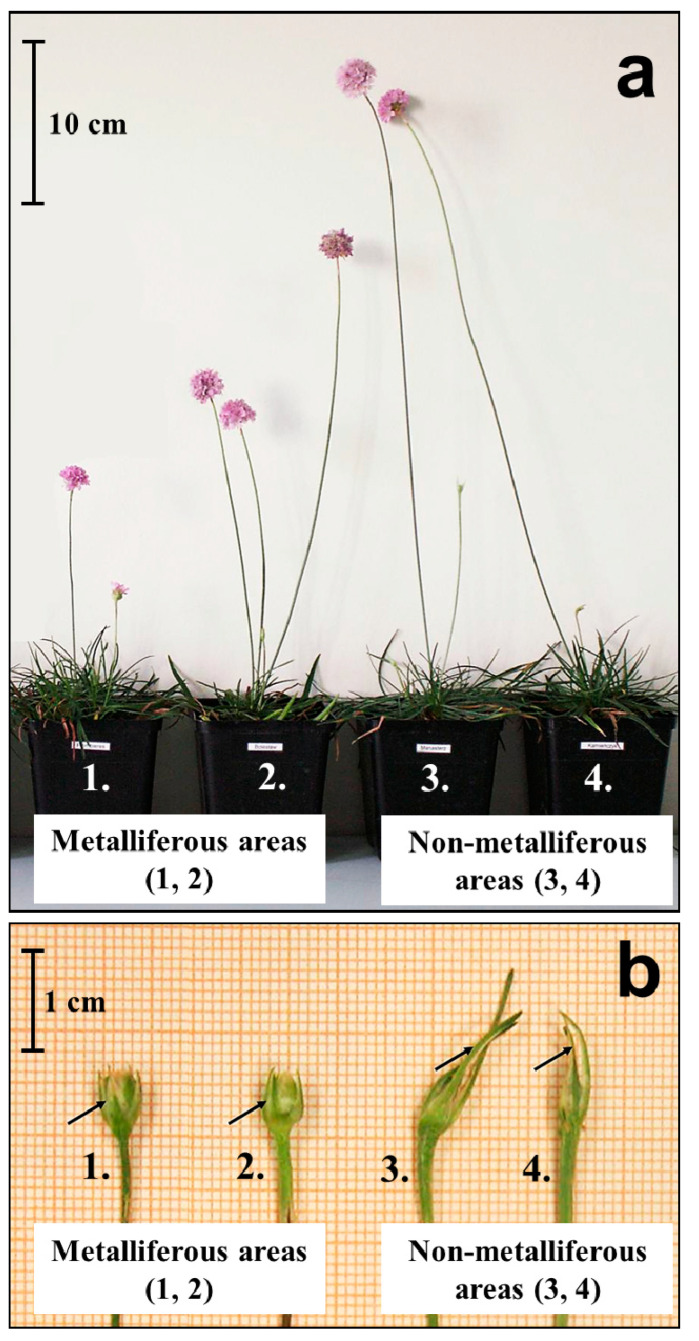
Morphological diversity of *A. maritima* plants grown in a greenhouse (generation F1) from seeds collected in metalliferous (1—zinc–lead slag heap from Plombières in Belgium; 2—zinc–lead waste heap from Bolesław in Poland) and non-metalliferous areas (3—meadow from northern Poland; 4—meadow from southern Poland). (**a**) The appearance of the whole plant. There are visible differences in the lengths of the generative shoots of plants from individual populations. Plants from metalliferous areas (1, 2) have shorter shoots than those from non-metalliferous areas (3, 4). (**b**) Inflorescence buds, where in plants from metalliferous areas (1, 2), the length of the outer involucral bracts of the inflorescence bud does not exceed the length of the bud (black arrows), while in plants from non-metalliferous areas (3, 4), it is twice as long as the bud (black arrows). All photos by Agnieszka Abratowska (based on Abratowska et al., 2015 [47], modified).

**Figure 4 ijms-24-04650-f004:**
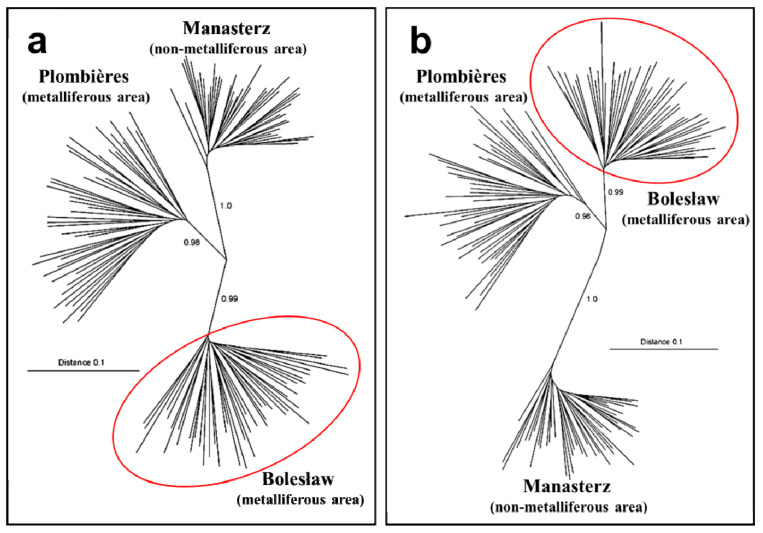
The unrooted neighbor-joining (NJ) trees based on AFLP data: (**a**) a *Kpn*I⁄*Mse*I methylation-insensitive enzyme combination and (**b**) *Acc*65I/*Mse*I methylation-sensitive enzyme combination from 136 individuals of *A. maritima* from two metalliferous populations—a zinc–lead waste heap from Bolesław in Poland (red oval) and a zinc–lead slag heap from Plombières in Belgium—as well as one non-metalliferous population—Manasterz in Poland (dry and semi-dry meadow). The numbers above branches denote bootstrap values, with Nei and Li’s genetic distances (based on Abratowska et al., 2012 [23], modified).

**Table 1 ijms-24-04650-t001:** Comparison of the appearance, morphology, and occurrence of three selected subspecies of *A. maritima* (based on: Szafer, 1946 [71]; Pinto da Silva, 1972 [69]; Lefèbvre, 1974 [76]; Woodell and Dale, 1993 [74]; Abratowska et al., 2012, 2015 [23,47]; Wąsowicz, 2015 [72]).

Features/Occurrence	Subspecies of *A. maritima*		
	Subsp. *Elongata*	Subsp. *Maritima*	Subsp. *Halleri*
Generative leafless shoot (scape)	A medium-sized plant with a scape that is glabrous or pubescent only at the base, 20–55 cm tall	A small plant with a scape that is densely pubescent, up to 5–25 cm tall	Usually a smaller, more delicately built plant than that of subsp. *elongata*; with a scape that is wholly glabrous, a little shorter, usually 10–25 cm tall
Leaves	Slightly acicular, up to 20 cm long, with the edge relatively densely and shortly ciliated at full length; leaves also pubescent on the nerve and the surface (usually in the lower part)	Usually 2–6 cm long, with the edge pubescent or glabrous; leaves generally thicker than in other subspecies	No longer than 8 cm long
Inflorescence	Outer involucral bracts of the inflorescence bud 7–25 mm long, green, leaf-shaped, lanceolate or ovate-lanceolate, usually elongated into a narrow and sharp tip, longer than inner involucral bracts; inner involucral bracts 5–10 mm long, rounded, blunt, or shortly pointed, the calyx tube pubescent only on the ribs; corolla usually light pink, although other shades of pink also possible	Outer involucral bracts of the inflorescence bud 4–5 mm long, ovate or elliptical, rounded, blunt or pointed at the tip, much shorter than inner involucral bracts; inner involucral bracts, usually 6–7 mm long; the calyx tube wholly pubescent or only on the ribs; corolla usually light pink	Outer involucral bracts are usually up to 4–5 mm long, broadly or narrowly ovate and shortly pointed, shorter than inner involucral bracts; inner involucral bracts 5–7 mm long and rounded at the tip; outer involucral bracts not higher than the inflorescence head, both in a bud and during flowering (in contrast to other subspecies in that the outer bracts may even be twice as high as a bud); the calyx tube pubescent on the ribs and between them; corolla usually dark pink/purple-pink
Occurrence	Common in the European Lowlands on sandy, unfertile, sunny grounds; dry and periodically wet meadows; fallow lands; roadsides; and forest edges	A halophyte occurring only in saline, mainly coastal areas, common in north-western Europe along the Atlantic coast to the western Baltic Sea coast	An absolute metallophyte associated with metalliferous areas with copper, zinc, and lead ores near mines and metal smelters
Plant community	A characteristic taxon of the *Diantho-Armerietum elongatae* association	A characteristic taxon of the *Armerion maritimae* association	A characteristic taxon of the *Armerion halleri* association and the *Armerietum halleri* (Libb. 1930) association

## Data Availability

The raw data supporting the conclusions of this article will be made available by the authors without undue reservation.

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
