# Peer review of "Micro-Evolutionary Processes in Armeria maritima at Metalliferous Sites"

_ijms, 2023, doi:10.3390/ijms24054650_

Round 1

Reviewer 1 Report

The author addressed the current knowledge of the heavy metal adaptation and genetic variation of A. maritima in zinc-lead waste heaps. The manuscript is interesting and relevant. In addition to this, the authors provided solid sentences and notions of heavy metal adaptation and genetic variation of A. maritima, which was in accordance with previous and current knowledge from published papers. Several comments are written below to be improved for the publication.

1. The title is not clear. The authors need to improve the title that contains more distinct meanings in the manuscript.

2. In line 40, subtitle of “2. Results” can be removed and the part is better to join into the part of “introduction”.

3. In line 122, considering that this review concentrates on the heavy metal adaptation as well as the genetic variation of A. maritima in zinc-lead waste heaps, the authors focused on the part of “Biology and ecology of America maritima” too much.

4. The part of “5. Avoidance…”, “6. A. maritima adaptation…”, and “7. A. maritima…” can be a subpart of 4 “Armeria maritima…”.

5. As well, the previous and current molecular mechanism of heavy metal avoidance, tolerance, and adaptations need to be discussed in the section.

Author Response

Response to Reviewer 1 Comments

Point 1: The title is not clear. The authors need to improve the title that contains more distinct meanings in the manuscript.

Response: The title of the manuscript has been changed to be more explicit. It is currently titled: “Microevolution processes in anthropogenic areas – Armeria maritima on a zinc-lead heap”.

Point 2: In line 40, subtitle of “2. Results” can be removed and the part is better to join into the part of “introduction”.

Response: In the second section of the paper (line: 40), the subtitle "Results" was removed and this part of the work has been included in the "Introduction" section.

Point 3: In line 122, considering that this review concentrates on the heavy metal adaptation as well as the genetic variation of A. maritima in zinc-lead waste heaps, the authors focused on the part of “Biology and ecology of America maritima” too much.

Response: In our opinion, it would be better if the review article included more basic information about the biology of Armeria mariitma species. Readers from different fields may find it useful.

Point 4: The part of “5. Avoidance…”, “6. A. maritima adaptation…”, and “7. A. maritima…” can be a subpart of 4 “Armeria maritima…”.

Response: According to the Reviewer's comment, the indicated subsections ("Avoidance and tolerance mechanisms", “A. maritima adaptation to heavy metals at the levels of the whole organism, individual tissues and cells" and "A. maritima adaptation to heavy metals at physiological and biochemical level") are now part of the section titled "Adaptations of Armeria maritima to heavy metals".

Point 5: As well, the previous and current molecular mechanism of heavy metal avoidance, tolerance, and adaptations need to be discussed in the section.

Response: The issue of the molecular basis of plant tolerance to heavy metals is a very broad topic, which is enough to write a separate review paper. Many articles have appeared in the literature on the above topic which are also cited in this work. In our manuscript, in the newly added final chapter with the title: "Future research", basic information on the subject has been added along with literature references. However, this type of research on A. maritima plants, has not yet been conducted. This is an excellent topic for the future.

Reviewer 2 Report

The review topic is interesting and the manuscript is well-written. Therefore, the manuscript has some problems that are listed below:

1) The novelty of the review is not clear. A high number of reviews have been published about plants and heavy metals; how does your review differ from them? Please, clarified in the last paragraph of the Introduction section.

2) The text is very good, but it is not clear with the information reported is from model conditions or from sites mentioned in Section 2. Please clarify it.

3) I suggest the authors synthesize the studies in a Table with the conditions tested and the consequences for the Armeria maritima. It will help the readers to find the information quickly and make the text more dynamic.

4) I also suggest a section of Future research to cite what information is lacking about this topic. It will help to guide future studies.

Author Response

Response to Reviewer 2 Comments

Point 1: The novelty of the review is not clear. A high number of reviews have been published about plants and heavy metals; how does your review differ from them? Please, clarified in the last paragraph of the Introduction section.

Response: The Introduction section was completed with an explanation of how the currently presented review differs from other review articles on plants and heavy metals (line: 120–122).

Point 2: The text is very good, but it is not clear with the information reported is from model conditions or from sites mentioned in Section 2. Please clarify it.

Response: This work has been completed with the information that indicates that the data presented in the section: "Adaptations of Armeria maritima to heavy metals" are from laboratory studies conducted under controlled conditions (line: 199–202)

Point 3: I suggest the authors synthesize the studies in a Table with the conditions tested and the consequences for the Armeria maritima. It will help the readers to find the information quickly and make the text more dynamic.

Response: The remark suggesting a synthesis of the studies presented in the table is very appropriate. However, different experimental variants were used in individual experiments. Such a table therefore seemed to us an oversimplification. On the other hand, at the end of the paper, in the "Conclusions" section, the most significant findings (which early on were presented in more detail in the manuscript) are listed in points.

Point 4: I also suggest a section of Future research to cite what information is lacking about this topic. It will help to guide future studies.

Response: In the manuscript, we have included a section titled "Future research”, which allows us to indicate the direction of future research.

Round 2

Reviewer 2 Report

The authors made the modifications and the quality of the manuscript increased.

Author Response

We thank you for your time and effort in reviewing our manuscript. The feedback has been invaluable in improving the content and presentation of the paper.